# Establishing an Ex Vivo Culture Model of Human Proximal Airway Tissue

**DOI:** 10.3390/mps8060132

**Published:** 2025-11-02

**Authors:** Neha Atale, Zihan Ling, Xi Ren, Kentaro Noda, Pablo G. Sanchez

**Affiliations:** 1Department of Pathology, University of Pittsburgh, Pittsburgh, PA 15213, USA; atalen2@upmc.edu; 2Department of Biomedical Engineering, Carnegie Mellon University, Pittsburgh, PA 15213, USA; zihanlin@andrew.cmu.edu (Z.L.); xir@andrew.cmu.edu (X.R.); 3Section of Transplant Surgery, Department of Surgery, University of Chicago, Chicago, IL 60637, USA; nodak@uchicago.edu; 4Section of Thoracic Surgery, Department of Surgery, University of Chicago, Chicago, IL 60637, USA

**Keywords:** human model, proximal airway, tissue culture model, protein synthesis

## Abstract

Background: Developing clinically relevant experimental models of the human airway can significantly advance our understanding of the mechanisms underlying airway diseases and aid in translating potential therapies to clinical settings. The aim of this study is to establish an ex vivo human airway tissue culture model. Methods: Human donor airway tissues were obtained from clinical cases of lung transplantation. Our established method is based on the concept of scavenging metabolic activity and controlling bacterial growth and includes increased media volume, frequent media exchange, and antifungal additives to efficiently maintain the homeostatic culture environment. After a 3-day culture period, the airway was investigated, and its viability and function were compared with a standard cell culture method. Results: Control tissue exhibited significant acidosis after 3 days, suggesting high metabolic activity of airway tissue and bacterial contamination. The airway epithelial viability—after culturing in our established method for 3 days—was better than that of the controls. We only performed an acute but early investigation of the cultures as airway complications have been known to start early at the proximal bronchus after transplantation. H&E and alcian blue staining showed intact morphology of the epithelium of airway tissue and mucus layers after 3 days in our model, while controls showed remarkable damage to the epithelial layer. Newly synthesized glycoproteins were detected in the epithelial layer using metabolic labeling and the click chemistry technique, suggesting cellular protein synthesis of the airway tissue in our established ex vivo model. Conclusions: We successfully established a reproducible model of human ex vivo airway tissue culture (n = 3 independent biological samples) that may be useful for investigating airway complications and developing their therapies.

## 1. Introduction

Developing clinically relevant experimental models of the human airway can significantly advance our understanding of the mechanisms underlying acute/chronic airway diseases and aid in translating potential therapies to clinical settings [1,2]. While primary cultures of human epithelial cells are commonly used as traditional experimental models, advanced models such as 3D-culture airway–liquid interface and ex vivo 3D lung tissue cultures such as precision-cut lung slices (PCLSs) have been established to investigate peripheral lung tissue [3,4,5,6]. Much research has been performed using the model of bronchus explant culture, where the peripheral bronchus explants are isolated from surgically resected tissue or organs, and a small piece of them can be incubated for up to 50 days [7]. A recent ex vivo model of large airway explants (LAEs) from unbranched bronchi also were shown to retain in vivo like properties for up to 14 days in culture. The authors showed that using Gelfoam as the substrate for the cultures prolonged the viability of the explanted tissue [8]. Researchers also cultured bronchial explants for up to 50 days to determine the lipid composition in the secretions [9]. The ex vivo models have great advantages over in vitro and in vivo studies when investigating toxicology, virology/infectious diseases, airway diseases (COPD/fibrosis), and biochemistry [9,10]. Bui et al. also showed the utility of explant cultures of human bronchus in testing for infections from the influenza B viruses (IBVs) and their replication in authentic human airway tissues. Bronchus explants preserve the complexity of airway epithelium, including ciliated cells, club cells, goblet cells, and basal cells [11]. On the other hand, explant modeling of distal bronchial cultures may not be sufficient to investigate clinical pathways of airway diseases such as extracellular remodeling, pathogenesis of epithelial damage, and subsequent epithelial recovery. Lung transplant outcomes commonly lead to bronchial anastomosis and complications related to it, which can significantly affect patient outcomes. The anastomosis is generally the most susceptible location associated with dehiscence, necrosis, stenosis, post-transplant bronchomalacia, fistula, and poor vascular circulation [12]. One of the common observations is early development of airway epithelial necrosis which specifically occurs at the intersection of the donor and recipient lungs due to poor blood supply leading to ischemia [13]. Bronchial dehiscence and stenosis are associated with high morbidity and mortality which are hard to study within in vivo conditions. Indeed, for our interests in studying airway complications in lung transplantation (e.g., epithelial necrosis, dehiscence, stenosis) that occur more frequently at the anastomosis of proximal bronchus, a culture model simulating the proximal airway using a large amount of tissue (at least 2–3 rings) would be necessary.

Following lung transplantation (LTx), the airway epithelium becomes particularly critical, as it is intricately involved in both the protection and vulnerability of the allograft. Airway epithelial cells contribute to innate immunity through pattern-recognition receptors, act as non-professional antigen-presenting cells, and produce immunomodulatory molecules. Given this dual role in defending against pathogens and regulating immune responses, the airway epithelium is central to many post-transplant complications, including infection, inflammation, and rejection. Therefore, understanding the functional state and immune activity of the airway epithelium and using an appropriate study model is essential for improving graft survival and long-term outcomes. In this regard, the intact culture of large airways with rich epithelial viability is desired and the model is established for such a specific focus. The challenges associated with ex vivo culturing of the human proximal airway include (1) sourcing intact samples of human tissue, (2) maintaining their metabolic activity, and (3) controlling bacterial growth. Human primary epithelial cells used in traditional culture studies are typically isolated from tissue biopsies or transbronchial swab samples [14,15]. Though, for ex vivo cultures of proximal airways such as the trachea and bronchus, alternative tissue sources for both normal and diseased airways are desired. Additionally, the high metabolic activity of airway tissue poses a significant challenge in maintaining homeostasis under ex vivo culture conditions [16,17], therefore the standard cell culture method is not suitable. Furthermore, bacterial growth is a common issue in primary human cell and tissue culture, and the airway represents one of the largest reservoirs of bacterial flora in the body, exacerbating this concern. Our proposed model is designed specifically to capture the immediate cellular and molecular events relevant to early airway complications. This focus is highly justified as early anastomotic failure and the initial inflammatory cascade leading to stenosis are critical events occurring within the first post-injury or post-transplant period. By utilizing a simple, static culture medium and a shorter incubation period, our model is significantly more cost-effective and resource-efficient than other protocols. Table 1 shows the comparison between our model with currently available different methods.

## 2. Materials and Methods

*Airway tissue culture and treatment*—Human donor proximal airway tissues (trachea and bronchus) were obtained from lung transplant cases with Institutional Review Board (IRB) approval (STUDY19080225) at the University of Pittsburgh as waste from OR cases. The samples were obtained from 3 patients. For each patient, we extracted trachea and bronchus. 3 tracheal rings were cut from each trachea. Duplicate observations for each sample were performed and averaged. The rings were selected randomly for culture in a control or optimized model. Tissues were washed with 1X PBS [Thermo Fisher Scientific Inc., Waltham, MA, USA, Gibco cat# 14-040-182] 3 times before culturing. For optimized culture conditions, a 1X antibiotic–antimycotic solution was added to the PBS used for tissue washing. The tissues were divided into two groups: (1) cultured in a 6 cm^2^ dish with 8 mL of medium (DMEM (with Bicarbonate) [Corning Inc., Corning, NY, USA. cat# 10-013-CV]/5%FBS [Gemini Bio, cat# 100-106]/10 mM HEPES [Thermo Fisher Scientific Inc., Waltham, MA, USA, Gibco cat# 15630080]/80 μL, 1X PenStrep [Thermo Fisher Scientific Inc., Waltham, MA, USA, Gibco, cat# 15-140-122]) with daily media change (control), [18,19,20,21,22] and (2) cultured in a 25 cm^2^ flask with 20 mL of medium (DMEM (with Bicarbonate)/5%FBS/20 mM HEPES/250 μL, 1X antibiotic–antimycotic [Thermo Fischer Scientific Inc., Waltham, MA, USA, cat# 15240062]) with twice-daily media change (optimized). In each case the medium engulfed the entire tissue sample. The tissue was cultured in each condition for up to 3 days at 37 °C in an incubator containing 5% CO_2_, and the tissue and culture media were analyzed to investigate their viability and cellular function. The average size and weights of the tracheal rings from 3 patients were 48.7, 67.4, and 72.3 mm, and 800, 920 and 1322 mg, respectively. Since we used different rings for control and optimized cultures for each patient, the size of the rings in the two conditions was the same.

*Triphenyl tetrazolium chloride (TTC) staining*—The human airway tissues before and during 3-day ex vivo culture was incubated in 0.1% of 2,3,5-Triphenyltetrazolium chloride (TTC) [MilliporeSigma, Burlington, MA, USA, cat# T8877-5G] in PBS for 1 h at 37 °C. TTC-stained tissue (~200 mg) was incubated in 1 mL of DMSO at 55 °C for up to 2 h to elute the stain. Absorbance of the eluted stain was measured at 480 nm wavelength using a microplate reader (BioTech Synergy HTX, Agilent Technologies, Santa Clara, CA, USA).

*Histological staining*—Airway tissue was fixed in 10% buffered formalin, embedded in paraffin (FFPE), and sectioned into 4 μm thickness. FFPE sections were deparaffinized by immersion in the xylene for up to 5 min; this was performed twice. Further rehydration was performed by passing it through different concentrations of alcohol such as 100, 95, 80, and 70% concentrations for 5 min, performed twice for each sample, and then passed through water. After that, slides were immersed in Hematoxylin for 3 min and rinsed with tap water. Counterstaining was performed by Eosin for up to 45 s, followed by bluing for 1 min. Dehydration was performed by increasing concentration of ethanol and slides were mounted with ClearMount^TM^ (Invitrogen, Waltham, MA, USA) after clearing with xylene. For alcian blue staining, slides underwent deparaffinization followed by rehydration and incubation in 3% acetic acid solution. Slides were then stained with alcian blue pH 2.5 at 37 °C for 1 h, washed, counterstained with Hematoxylin, and mounted for scanning. For TUNEL staining, airway tissues were stained with Click-iT plus TUNEL assay-Alexa Fluor 647 dye (Invitrogen, Waltham, MA, USA) according to manufacturer’s protocol for apoptosis detection. After the 3-day culture period, both control and optimized tissues were harvested and counterstained with Hoechst 33,342 staining. All stained sections were scanned using a whole-slide image scanner (Axio Scan.Z1; Zeiss AG, Oberkochen, Germany) and analysed with digital image processing software (ZEN lite blue edition 2012 version 1.1.2.0; Zeiss AG, Oberkochen, Germany).

*Metabolic labeling and click chemistry*—Metabolic labeling was performed overnight for the tissue in our established culture model at day-3 using tetraacetylated N-Azidoacetylgalactosamine (Ac4GalNAz) probe as performed previously [23]. Ac4GalNAz probe concentrations of 0, 50, and 100 μM were tested. The labeled tissues were homogenized in CHAPS buffer. Affinity purification was performed by click chemistry with copper-catalyzed azide–alkyne cycloadditions (CuAAC) and the extracts reacted to Alkyne-PEG4-biotin under a copper I ion existence. Western blotting for the extracted protein was performed using horseradish peroxidase (HRP)-conjugated streptavidin (Thermo Fisher Scientific, Waltham, MA, USA) at a 1:10,000 dilution for 1 h, conducted following gel run and transfer. For immunostaining of Azide incorporation in airway tissue, the tissue after metabolic labeling with 100 μM Ac4GalNAz was fixed in 10% buffered formalin, embedded in paraffin, and sectioned into 4 μm thick pieces. The deparaffinized sections were incubated with primary antibody against Laminin (Abcam, Cambridge, UK. cat# ab11575, dilution 1:500). Furthermore, slides were scanned after staining with streptavidin conjugated to Alexa Fluor-647 (Thermo Fisher Scientific, Waltham, MA, USA) and Donkey-anti-Rabbit IgG conjugated to Alexa Fluor-488 (both at dilution 1:500, Thermo Fisher Scientific, Waltham, MA, USA).

*Statistical analysis*—Statistical analysis was performed using GraphPad Prism software (GraphPad Software, Boston, MA, USA, Ver.10.). The results were expressed as mean ± standard deviation. Significance between means was determined using a non-parametric Wilcoxon matched-pairs signed rank test, with *p* < 0.05 considered statistically significant in all experiments.

## 3. Results

We examined epithelial viability and functionality of the human airway using the optimized ex vivo culture model.

Under the approval of the IRB, we have established a pipeline for obtaining primary proximal airway tissue from donor lungs obtained during lung transplant procedures, which are typically discarded as operating room waste. Utilizing this tissue, we have developed an optimized culture protocol for culturing the proximal airway. To address the challenges mentioned previously, our optimized protocol involves several key steps. Firstly, the airway tissue undergoes a pre-culture wash with a bulk volume of antibacterial saline solution. This step aims to reduce bacterial contamination. Secondly, we perform twice-daily media exchanges using a larger volume of medium. This facilitates the removal of metabolites and provides essential nutrients to maintain tissue homeostasis. Finally, we supplement the culture medium with antibiotic–antimycotic agents to further inhibit bacterial and fungal growth. We also increase the concentration of HEPES, a buffering agent renowned for its capacity to maintain physiological pH levels, within the culture medium containing airway tissue. This adjustment is undertaken to ensure the maintenance of pH within the medium, thereby facilitating optimal conditions for airway tissue culture. These modifications are designed to mitigate bacterial and fungal contamination, ensure culture homeostasis, and efficiently scavenge metabolite accumulation. In the control cultures, the rapid progression of acidosis in the culture media (~5.0) due to high metabolic requirements was observed. This was accompanied by a change in color of media to a pale yellow and exhibited turbidity, suggesting excessive bacterial growth. The pH measurements revealed acidic pH in the control culture containing 10 mM HEPES (~5.0), while 20 mM HEPES containing media from optimized culture maintained the pH up to the physiological level (~7.0). Our protocol here differs significantly from many peripheral bronchus explant culture protocols that do not add HEPES to control pH levels [8,22].

Airway tissue was collected from the same tissue of the same patient and cultured for up to 3 days and in two groups: (1) 6 cm^2^ dish with 8 mL medium (DMEM/5% FBS/10 mM HEPES/PenStrep) with daily media change (control) and (2) 25 cm^2^ flask with 20 mL medium (DMEM/5% FBS/20 mM HEPES/antibiotic–antimycotic) with twice-daily media change (optimized) as indicated in the schematic design in Figure 1A. To confirm the effectiveness of our optimized protocol for ex vivo tissue culture of the proximal airway, we investigated the tissue viability and functionality of the airway after 3 days of culture and compared them with the results from the standard cell culture method. The epithelial viability was confirmed through their mitochondrial activity using TTC staining and the epithelial layer of the proximal airway after the 3-day control culture showed a lack of TTC staining. A direct connection between medium acidosis and cell viability has been well established [24]. High to medium acidosis leads to lower cell viability [25,26]. Our optimized culture prevented the progression of medium acidosis, quantified by pH measurements and color of the medium. This resulted in a significantly higher viability of the epithelial layer after the 3-day culture period, compared to that of pre-culture and control samples. We extracted the TTC stain in tissue with DMSO to quantitate its intensity and confirmed a significant enhancement in viability in the optimized tissue compared to that of the controls over the course of 3 days (*p* < 0.003, Figure 1B,C. Individual data points plotted in Appendix A). The cell viability for the control decreases due to increasing damage in the mitochondria with time and the increase in TTC labeling in optimized culture leading to increases in the active mitochondria and viability of tissue.

H&E staining revealed preserved morphology of epithelial layer in tissues after 3-day culture with the optimized protocol, while it was disrupted in controls (Figure 2A). To confirm the function of airway epithelium, we performed alcian blue staining for the cultured tissue, which showed intact mucus layers secreted by goblet cells and submucosal glands in tissues after a 3-day culture period with the optimized protocol. However, it did not show either mucosal layer or submucosal glands in control tissues (Figure 2B, images for larger tissue sections for each patient and culture condition are shown in Appendix A). The integrity of the mucus layer and the presence of goblet cells and submucosal glands are key indicators of epithelial function. We examined the cell death in the airway tissue after each culture condition by TUNEL staining. A higher number of TUNEL-positive cells were found in the epithelium in the 3-day cultured tissue of the control group, whereas tissue after 3-day optimized culture showed significantly fewer positive cells in the epithelial layer (Figure 2C. TUNEL images for larger tissue sections for two patients are shown in Appendix A). These findings indicate that our optimized culture restored the epithelium structure and viability.

We aimed to assess whether the culture optimization protocol could preserve protein synthesis function in airway tissue. To investigate this, we performed metabolic labeling using a tetraacetylated N-Azidoacetylgalactosamine (Ac4GalNAz) probe at a concentration of 100 μM, as previously described [23], on day 3 of ex vivo cultures overnight. Azide labeling of newly synthesized glycoproteins (NewS) in airway tissue were employed to precisely identify and analyze these proteins in the epithelium through the alkyne-biotin conjugation process, facilitated by copper-catalyzed cycloaddition conditions. This assay was conducted to assess epithelial function and was therefore not performed on control samples, which lacked an intact epithelial layer. This method offers an efficient tool for studying the specific interactions and roles of NewS glycoproteins within the airway microenvironment (Figure 3A). Western blot analysis of the azide-biotin conjugation revealed a dose-dependent increase in signal intensity of the extracted labeled proteins from the tissue (Figure 3B). Immunofluorescence staining demonstrated robust protein labeling in the epithelial layer of the optimized 3-day culture, indicating active protein synthesis. In contrast, the negative control did not show any labeling (Figure 3C). These results collectively demonstrate that our established ex vivo model preserves cellular viability, physiological morphology, functional mucous membrane, and metabolic activity, making it a suitable model for studying airway and lung pathologies.

## 4. Discussion

To date, the available 2D experimental models using human primary cells or tissues have limitations in investigating crucial aspects of airway biology, such as cell–cell interactions within the actual airway structure, extracellular matrix (ECM) interactions, epithelial turnover, and repair mechanism. For example, understanding airway epithelial repair, particularly the behavior and mechanisms of progenitor cells (e.g., basal cells), remains a significant challenge [27]. Our ex vivo model allows us to address these limitations and provides a versatile platform for investigating these critical topics. By combining this model with advanced research tools commonly used in 2D in vitro models, such as real-time multi-photon microscopy and bioengineering approaches, we can study various acute airway injuries, including aspiration, hypoxia, and infectious diseases. This allows for real-time observation of cellular behavior, healing processes, metabolic alterations, immune cell interactions, gene transfection/editing, and testing of potential therapies. While our initial study utilized normal proximal airway tissue from human donors to establish the ex vivo model, the potential applications of this model could be further expanded by using diseased bronchial tissue from lung transplant recipients. This would enable investigation into end-stage lung diseases such as idiopathic pulmonary fibrosis (IPF), scleroderma, gastroesophageal reflux disease-associated lung injury, and asthma. In lung transplant settings, where airway anastomosis connects the healthy donor’s proximal airway with the recipient’s diseased one, complications such as epithelial necrosis/dehiscence and stenosis remain significant issues affecting patient prognosis [28]. The mechanisms underlying post-transplant airway complications, including their end-to-end communication between donor and recipient airways, ischemia due to lack of bronchial artery perfusion, immunological interactions, [29] and ECM degradation, have not been thoroughly investigated. Mitomycin C (MMC) is applied topically following surgical intervention to reduce scar tissue formation and delay restenosis. Our model is useful for investigating the effects of different therapies such as the use of MMC to prevent airway stenosis [30]. Our ex vivo model has the potential to elucidate these mechanisms and signaling pathways, thus contributing to a better understanding of these conditions and the development of novel therapeutic approaches.

Platelet-rich plasma (PRP) has also gained attention in tissue repair and regenerative medicine, including treatment of complex fistulas [31]. In preclinical studies, including large animal models, PRP has been shown to facilitate anastomotic healing following tracheal resection. Enhanced revascularization and tissue integration at the surgical site have been attributed to the local release of platelet-derived growth factors (PDGFs), further supporting PRP’s biological activity in complex airway repair [32]. These findings provide a compelling rationale for incorporating PRP into ex vivo systems, where the mechanisms of PRP-mediated healing can be further dissected under controlled experimental conditions. Our ex vivo model offers a unique platform to study the localized effects of PRP on airway tissue, enabling real-time observation of tissue remodeling and inflammatory responses in a physiologically relevant context.

Finally, our ex vivo airway culture model offers a simple yet effective approach for investigating airway biology compared to other advanced research models, such as 2D or 3D models. For investigating end-stage lung disease, our group previously introduced the ex vivo lung perfusion model [33], and PCLS is also considered as alternative ex vivo model [3]. However, these models require technical maturation and entail higher costs. EVLP involves a complex warm autopsy procedure to obtain diseased lungs [33]. In contrast, our model utilizes a simplified setup and a modified culture protocol, utilizing samples typically discarded as operating room waste in clinical lung transplantation cases. These findings suggest that this model is particularly suitable for investigations involving extended culture durations (up to 72 h), as tissue changes became more pronounced with prolonged incubation.

While our model offers a valuable platform to study airway complications, it does have several limitations. Our study was limited to only 72 h compared to other specialized protocols that are carried out for up to 14–50 days. In contrast to these studies, our aim was to study acute responses in the donor tissues, as the EVLP for lung transplants are usually only required for this duration. Further, it is well known that airway complications are triggered very early at the proximal bronchus junctions and take effect within the first 72 h. Furthermore, our model also lacks an analysis of functional properties of the cultures such as airflow, ciliary motion, and shear stress which reduces its physiological relevance, particularly in studies involving mucociliary clearance or epithelial barrier integrity. Future work could incorporate assays such as transepithelial electrical resistance (TEER) or paracellular flux measurements to gauge barrier integrity, ELISA or other methods to quantify the release of relevant cytokines and inflammatory mediators, and high-speed video microscopy to evaluate ciliary beat frequency (CBF) and overall ciliary motion. The scalability of the model is also limited, as access to healthy human bronchial tissue is challenging, and donor-to-donor variability introduces additional concerns regarding reproducibility. Furthermore, the viability of explanted tissue is typically short-lived, which confines the model to acute studies and hinders exploration of long-term processes such as epithelial remodeling, chronic inflammation, or rejection. The current model lacks dynamic mechanical cues such as the cyclic stretch and compression that occur with normal breathing, as well as the airflow and associated shear stress across the luminal surface. These mechanical forces are known to be critical regulators of epithelial cell function, including differentiation, barrier integrity, and ciliary beat frequency, and their absence may lead to non-physiological cellular responses or maintenance of phenotype. Furthermore, the model inherently omits the complexities of the systemic immune and vascular compartments. Crucially, the absence of circulating immune cells (e.g., T cells, B cells) and the local immune microenvironment means that our study cannot evaluate host–graft immunological interactions, the potential for allograft rejection or chimerism, and the full spectrum of inflammatory responses that would occur in vivo. Although by applying our simpler ex vivo airway culture model, we can push the boundaries of therapeutic development in bioengineering, regenerative medicine, and pharmacology for acute and chronic airway diseases. This model holds promise for accelerating the translation of experimental findings into clinical applications, ultimately benefiting patients with airway diseases.

## Figures and Tables

**Figure 1 mps-08-00132-f001:**
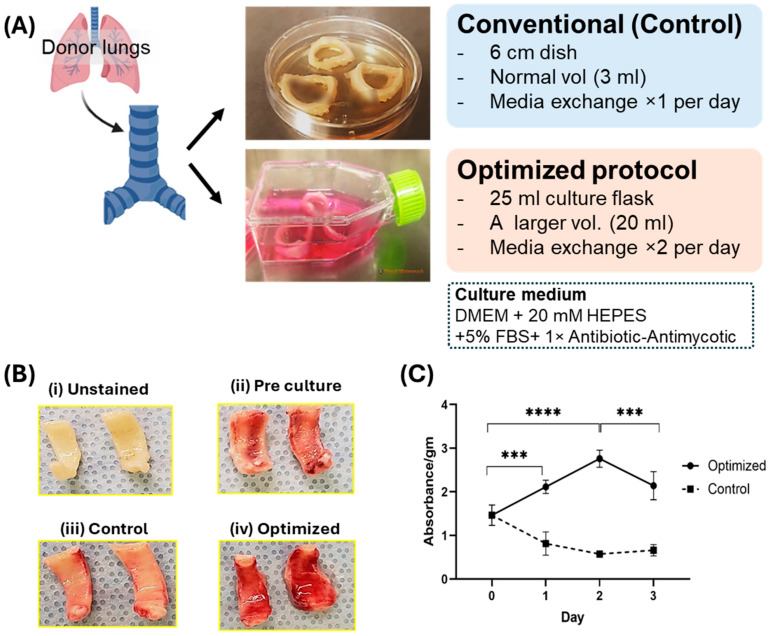
**Establishment of ex vivo culture airway model**. (**A**) Study design: Schematic presentation of the design of our model, indicating the culturing of the airway using the conventional and optimized method. (**B**) Comparative analysis of TTC staining. Figure revealed differential staining intensity in four distinct groups: (i) Unstained, serves as a negative control (without staining); (ii) Pre-culture, represents tissues immediately after cold ischemia; (iii) Control, indicates tissue subjected to conventional culturing conditions; and (iv) Optimized, denotes tissues treated under optimized culturing conditions for 3 days. Remarkably, the optimized group demonstrates significantly enhanced positive TTC staining, indicative of epithelial viability, in comparison to the pre-culture condition. The control group does not exhibit positive staining, denoting the loss of epithelial viability. (**C**) Quantitation of TTC stained tissue. Quantitative data for TTC reveals high positive staining in optimized tissue (n = 3, independent biological samples) after 3-day culture as compared to both pre-culture condition and control (*p* < 0.003). ***: *p* < 0.003, ****: *p* < 0.001.

**Figure 2 mps-08-00132-f002:**
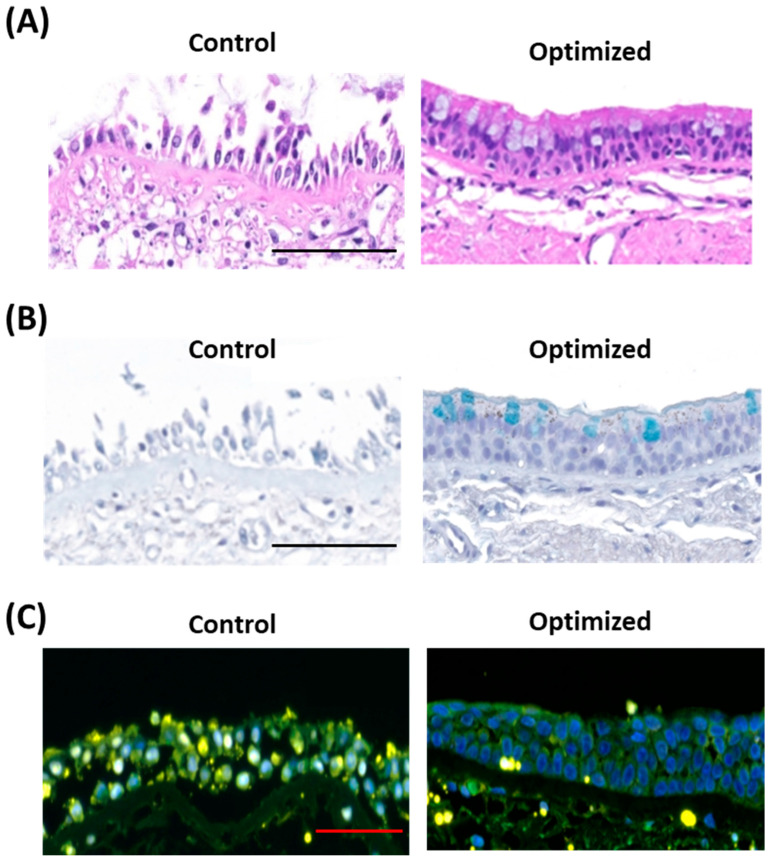
**Epithelial viability and function in ex vivo airway culture model.** (**A**) Hematoxylin and Eosin (H&E) staining of epithelium: Cross-sectional images of the airway epithelium reveals preserved epithelial morphology in the optimized culture, whereas significant epithelial damage is observed in the control tissue. Scale bar: 100 μm. (**B**) Alcian blue staining of mucus layer: The optimized culture maintains a well-preserved mucus layer, while the control exhibits disruption of the mucus layer and absence of goblet cells. Scale bar: 100 μm. (**C**) TUNEL assay: The control group displays a high frequency of TUNEL-positive cells within the Hoechst-stained epithelial layer, indicative of substantial apoptotic activity. In contrast, the optimized culture exhibits intact epithelial layers with an absence of TUNEL-positive cells, suggesting minimal apoptosis. Scale bar: 50 μm. Images of the larger sections of the tissue are provided in Appendix A.

**Figure 3 mps-08-00132-f003:**
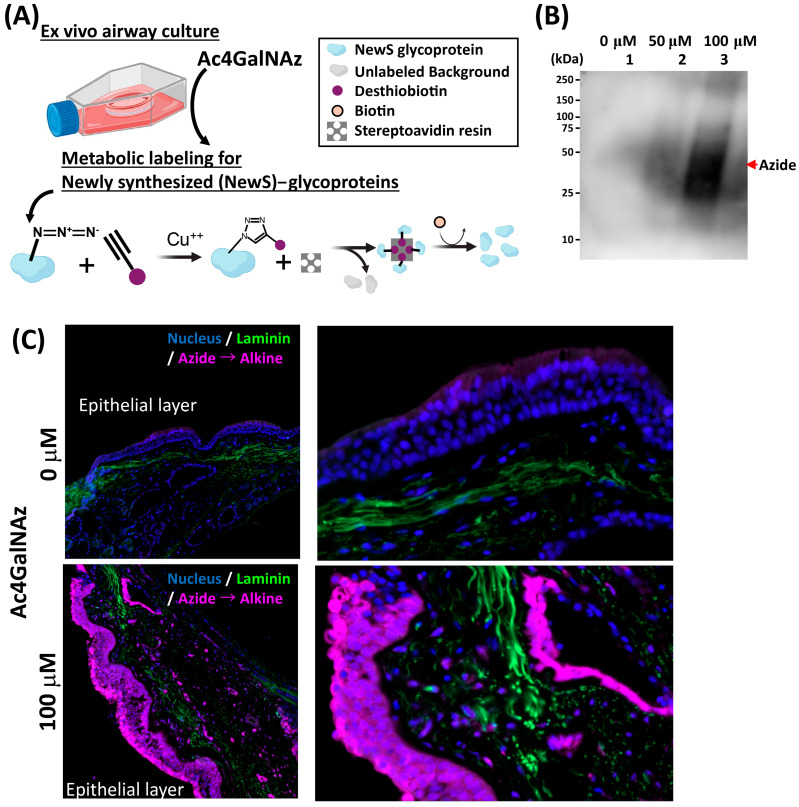
**Metabolic labeling of NewS glycoproteins in ex vivo airway culture.** (**A**) Schematics of labeling of newly synthesized (NewS) glycoproteins. Metabolic labeling of NewS glycoproteins was carried out using a 100 μM concentration of tetraacetylated N-Azidoacetylgalactosamine (Ac4GalNAz) probe, which was added to the culture on day 2 and incubated overnight. During labeling, extracts containing the target glycoproteins were treated with Alkyne-PEG4-biotin in the presence of copper(I) ions, promoting the covalent attachment of the biotin moiety to the glycoproteins through a copper-catalyzed azide–alkyne cycloaddition reaction. The biotinylated proteins were then selectively captured using streptavidin and allowing for their isolation and subsequent analysis as indicated by chemical reaction. (**B**) Western blot analysis: Dose-dependent increase in the probe correlated with the enhanced signal intensity of extracted labeled proteins in tissue. (**C**) Immunofluorescence staining for NewS glycoprotein labeling. Successful molecular probe (Azide) labeling newly synthesized glycoproteins during the designated 18 h period of culture. In contrast, the absence of labeling was observed in negative control.

**Table 1 mps-08-00132-t001:** Comparison of current methods for establishing lung explant models with our proposed model.

	Precision-Cut Lung Slices (PCLSs)	Air–Liquid Interface (ALI)	Ex Vivo Lung Perfusion (EVLP)	Short-Term 3-Day Trachea Culture (Proposed Model)
Model	Thin tissue slices ranging from 150 to 600 μm	3D Cell culture	Whole-organ system which maintains the entire lung under near-physiological conditions (normothermia, perfusion)	Intact tracheal rings
Airway used	Alveoli and small airways/bronchioles, or large airways	Primarily bronchial/tracheal epithelial cells	Entire donor lung	Mainly trachea and bronchus
Duration	3–5 days	Long term, 1 month	Short term, 4–12 h	Acute, up to 3 days
Fidelity	High	Medium	Highest	High
Cost	High	Medium	Highest	Low
Throughput	High throughput		Very low throughput	Medium
Outcome	Limited long-term viability	Pure cell culture	Cost, complexity, requirement of well-equipped perfusion platform	Suitable for short-term assessment, no cost complexity
Function	Provides a viable 3-dimensional lung tissue model	Mimic in vivo environment of airway by promoting pseudostratified morphology, mucociliary differentiation (cilia and mucus).	EVLP enhances the performance of donor lungs, thereby expanding the pool of transplantable organs and helping decrease mortality among patients on the transplant waiting list	Appropriate model to study acute phase of stenosis and anastomosis in proximal airways

## Data Availability

The original contributions presented in this study are included in the article/Appendix A. Further inquiries can be directed to the corresponding author.

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
