# Peer review of "Establishing an Ex Vivo Culture Model of Human Proximal Airway Tissue"

_mps, 2025, doi:10.3390/mps8060132_

Round 1

Reviewer 1 Report

Comments and Suggestions for Authors

The paper „Establishing an ex-vivo culture model of human proximal airway tissue” by Neha Atale et al. gives new insights into ex-vivo cell culture of airway tissue. The introduction is well-written, only the term “biochemistry” could be more specific by adding some processes that need to be investigated. The methods are clear and it would be relatively easy to follow and repeat them. The obtained results are discussed in a nice way including limitations of the study. It is not known for how long the tissues could be cultured to mimic chronic conditions but the authors refer to this problem. There is one thing that could be added and maybe discussed – the causes of lung transplants in tissue donors and whether the main reason/cause/disease could influence the outcomes of the study. Fig 3B should be corrected.

Otherwise, this is a very nice article worth publishing and gaining with time more and more attention.

Author Response

Comment 1. There is one thing that could be added and maybe discussed – the causes of lung transplants in tissue donors and whether the main reason/cause/disease could influence the outcomes of the study.

Answer 1: Thank you very much for this question. The source of the tissue is the donor lung grafts. The airway tissue of organ donor lungs is usually trimmed, and the proximal airway (trachea, carina, and main bronchus) is discarded at transplant surgery. Those tissues should be healthy therefore they are considered great resource to study normal airway for tissue culture/engineering since it is difficult to take biopsy of such samples from healthy individuals.

Comment 2. Fig 3B should be corrected.

Answer 2: Thank you for the suggestion. We have already submitted the corrected figure as per our understanding.

Reviewer 2 Report

Comments and Suggestions for Authors

The study focused on establishing an ex-vivo culture model of human proximal airway tissue. In particular the author provided a protocol that maintain viable and intact airway epithelium for three days in a submerged, ex vivo culture.

Major concerns:

While the described protocol may have advantage from its simplicity, it has significant limitation in its application due to non air-liquid interface condition as well as short culture period. Through the introduction and the discussion, the authors described advantage and possible application of “general” ex vivo graft model. This strongly misleads that the authors were trying to establish a model that could be used for the mentioned applications, and the optimized culture model described in this study was such. In addition, although the authors mentioned some limitations and studies that may not be applied in discussion, in elsewhere, they claimed that the model can be used for many studies required ALI condition and long-term culture.

The authors seem to have a specific application and targeted airway physiology to study in mind (perhaps lung transplantation outcome). I believe it becomes more compelling story if the authors had built the manuscript based on it and described that the submerged culture could suffice their needs (or they do not need complicated set up, ALI, and/or long-term culture) and its advantage for the specific application (large graft with high viability, etc).

This directional change (removing generality) may help to address another major concern regarding to the control condition used. While the author showed significant improvement in viability of epithelial cells in ex-vivo graft using an optimized condition when compared to a condition that they referred “conventional” control, the control does not appear appropriate. The given references for the control culture conditions are for cell lines and/or other tissues, but not airway graft culture. Thus, it is unclear the advantage of the method described here compared to the ones used in the field to maintain airway tissue ex vivo. In addition to the authors reference (Ref 10), there are several more studies establishing human and non-human large airway and trachea graft at air-liquid interface culture for extended period.

The following are additional comments and critiques:

Major comments:

  1. Abstract: the author claims their “optimized” protocol is a reproducible model, however it is unclear how experiments were reproduced. See further details in below.
  2. Introduction: lines 7-14, lung transplant outcome …, are disrupting the story of modeling before and after these lines. The authors may want to move these to the end of the paragraph describing author’s targeted area, needs of this study, the reason of establishing an ex-vivo model in the manner that authors did, etc.
  3. Lines 27-30, on the other hand, ….: if this is due to the quantity, please specify so within the same sentence. Otherwise, this sentence is conflicting with advantage of ex vivo culture.
  4. Material and method: the authors mentioned this ex-vivo model were established with/optimized for trachea and bronchus, but only trachea seems to be used based on the method in this study. Please clarify.
  5. Material and method: it is unclear why conventional “cell” culture method is the control of this study.
  6. Results, last sentence of first paragraph: please provide references. Might these conditions be more appropriate control in this study despite it is for small airway?
  7. Results, second paragraph and fig. 1C: Please clarify how the experiments were repeated (or multiplicated) and the meaning of n=3. 3 donors (biological replicates) data should be presented as the authors described in materials and methods.
  8. Results, third paragraph and fig. 2A-C: Please provide macro-image from stitching/tile-scanning to confirm that the shown images were not selected. It is necessary and common in the field. Also please show all three donors’ data for reproducibility purpose. These data may go to supplemental figures.
  9. Discussion, 3rd paragraph: extended culture durations (>72 hours) is misleading. The authors tested up to 72hrs, thus it should be extended culture durations (up to 72 hours).

Minor comments:

  1. Material and method: since this is an article focused on a method, the authors should include detailed information of used reagents for reproducibility. In particular, vendor and catalog number should be included for the key materials such as cell culture reagents and antibodies.
  2. Material and method: there is a discrepancy in volume of medium in the optimized condition. There are 20ml and 25ml. Please correct.

Author Response

Comment 1. Abstract: the author claims their “optimized” protocol is a reproducible model, however it is unclear how experiments were reproduced. See further details in below.

Answer 1. Thank you very much for pointing out this. We have edited the abstract to mention the n number.

Changes made: We added that 3 independent samples were used to reproduce the results [Line 31]

We successfully established a reproducible model of human ex-vivo airway culture (n=3 independent biological samples) that may be useful for investigating airway complications and developing therapies.

Comment 2. Introduction: lines 7-14, lung transplant outcome …, are disrupting the story of modeling before and after these lines. The authors may want to move these to the end of the paragraph describing author’s targeted area, needs of this study, the reason of establishing an ex-vivo model in the manner that authors did, etc.

Answer 2. Thank you very much for suggestion. We have moved the introduction lines to the end of the paragraph so that the introduction talks about limitations of distal bronchus cultures and the requirement of doing proximal airway cultures [Line 66-72].

The airway epithelial necrosis is often observed after lung transplantation and associated with dehiscence or stenosis if it goes severe or abnormal recovery occurs. After anastomosis of donor airway to the recipient airway in lung transplantation, the dnor airway does not receive any blood perfusion because it takes time for bronchial circulation being reestablished. Thus such ischemic condition in airway leads to epithelial necrosis and we have interest in establishing donor airway preservation methods and prevention of airway ischemia epithelial necrosis in lung transplantation.

Changes made: We have revised the text as per the suggestion [Line 66-72].

Comment 3. Lines 27-30, on the other hand, ….: if this is due to the quantity, please specify so within the same sentence. Otherwise, this sentence is conflicting with advantage of ex vivo culture.

Answer 3. We are referring here to the ex vivo cultures of peripheral bronchus specifically and the fact that cultures of proximal airways better model the pathogenesis of the lung transplant complications. [please see Line 63-64].

Changes made: We have corrected the text to rephrase this line [Line 63-64].

On the other hand, explant modelling of distal bronchial cultures may not be sufficient to investigate clinical pathways of airway disease such as extracellular remodeling, pathogenesis of epithelial damage and subsequent epithelial recovery

Comment 4. Material and method: the authors mentioned this ex-vivo model were established with/optimized for trachea and bronchus, but only trachea seems to be used based on the method in this study. Please clarify.

Answer 4. Thank you for mentioning this. While we extracted both bronchus and trachea from the patients, only tracheal rings for culture were used.

Comment 5. Material and method: it is unclear why conventional “cell” culture method is the control of this study.

Answer 5. Thank you for your point. We mentioned the traditional cell culture method for reference as it is closest available protocol we have used but traditionally similar protocol is used for establishing tracheal explants.

Changes made: We have added two more references on tracheal explants protocols that follow the same methods. [Line 126]

  1. Gabridge MG, Bright MJ, Agee CC, Nickerson JM, and Henderson NS. Development of an improved tracheal explant bioassay for the detection of the ciliary dyskinesia factor in cystic fibrosis serum. Pediatr Res 13: 31-35, 1979.
  2. Abeynaike L, Meeusen EN, and Bischof RJ. An ovine tracheal explant culture model for allergic airway inflammation. J Inflamm (Lond) 7: 46, 2010.

Comment 6. Results, last sentence of first paragraph: please provide references. Might these conditions be more appropriate control in this study despite it is for small airway?

Author response: Thank you for bringing this to our notice. We have mentioned corresponding references 10 and 22 in the manuscript.

Changes made: We have mentioned the reference in the text [Line 206].

Our protocol here differs significantly from many peripheral bronchus explant culture protocols that do not add HEPES to control pH levels (10, 22).

Comment 7. Results, second paragraph and fig. 1C: Please clarify how the experiments were repeated (or multiplicated) and the meaning of n=3. 3 donors (biological replicates) data should be presented as the authors described in materials and methods.

Answer 7. Thank you for the query. The experiments were repeated on 3 independent samples from patients. We have added the meaning of n. (n=3, independent patient samples).

Changes made: We have made the change to the figure legend and also provided the data points for each patient as supplementary figure 1.

Quantitative data for TTC revealed high positive staining in optimized tissue (n=3, independent biological samples) after 3-day culture as compared to both pre-culture condition and control (p<0.003).  ***: p<0.003, ****: p<0.001.

Comment 8. Results, third paragraph and fig. 2A-C: Please provide macro-image from stitching/tile-scanning to confirm that the shown images were not selected. It is necessary and common in the field. Also please show all three donors’ data for reproducibility purpose. These data may go to supplemental figures.

Answer 8. Thank you for the comment. We include the images with a low magnification in supplementary figure 2 and 3.

Changes made: We have added the H&E, AlcianB and TUNEL images in Supplementary Material with a lower magnification to visualize the whole section (~2 mm).

Comment 9. Discussion, 3rd paragraph: extended culture durations (>72 hours) is misleading. The authors tested up to 72hrs, thus it should be extended culture durations (up to 72 hours).

Answer 9. Thank you for pointing out. We revised the text accordingly.

Changes made: We have made correction in this line “These findings suggest that this model is particularly suitable for investigations involving extended culture durations (up to 72 hours), as tissue changes became more pronounced with prolonged incubation [Line 312-313]”.

Minor comments:

Comment 1. Material and method: since this is an article focused on a method, the authors should include detailed information of used reagents for reproducibility. In particular, vendor and catalog number should be included for the key materials such as cell culture reagents and antibodies.

Answer 1. Thank you for pointing this out. We have added additional details of the reagents and materials used in the methods.

Changes made: We have added these details in the material and methods section. [Lines 120 – 130].

Comment 2. Material and method: there is a discrepancy in volume of medium in the optimized condition. There are 20ml and 25ml. Please correct.

Author Response: Thanks for a careful review and finding our typo. We have corrected the figure to 20 mL.

Reviewer 3 Report

Comments and Suggestions for Authors

The manuscript reports an ex-vivo culture model of human proximal airway tissue sourced from discarded donor material during lung transplantation. Optimisations, larger medium volume, more frequent exchanges, antifungal/antibiotic supplementation, and increased HEPES buffering, preserve epithelial viability and function for at least three days. This approach fills a gap between simple in-vitro epithelial systems and complex, costly ex-vivo lung platforms that skew towards distal/peripheral tissues. The proximal focus is clinically relevant for post-transplant complications. Related models exist (PCLS, ALI, bronchial explants; e.g., Snyder 1984; Jozwiak 1984; Lee-Ferris 2025), some supporting longer viability, but these are largely distal/peripheral. The present protocol emphasises proximal bronchus/trachea, metabolic homeostasis, and contamination control; its novelty is simplicity, reproducibility, and proximal emphasis. Below, I highlight some points I think can be improved. 

Major Revisions

  1. The authors should more explicitly distinguish their model from recent large airway explant cultures (Lee-Ferris et al., 2025) that also demonstrated viability up to 14 days. Current framing risks overstating novelty.

  2. Suggest adding a comparative table summarising existing models (ALI, PCLS, explants, EVLP) versus the proposed protocol.

  3. The study only reports up to 3 days of culture, while earlier models showed extended viability. This limitation must be clearly acknowledged in the abstract and discussion, with justification for the chosen endpoint.

  4. Only three donor samples were used. This small number raises concerns about reproducibility and donor variability. The limitation is mentioned but needs stronger emphasis, ideally with data on inter-donor variability.

  5. Beyond glycoprotein synthesis, functional assays (e.g., barrier integrity, cytokine release, ciliary motion) would strengthen the model’s translational value. The absence of such data should be discussed.

  6. The authors suggest use in drug testing and translational research, but with only short-term viability shown, this claim seems overstated. Recommend tempering claims or providing preliminary pilot data (e.g., exposure to therapeutic agents).

Minor Revisions

  1. Revise phrasing in the abstract: “Control tissue exhibited significant acidosis…” should clarify whether this is compared to pre-culture baseline.

  2. Include explicit mention of study limitations (short viability, small sample size) in the abstract and introduction.

  3. Provide more details on medium composition (e.g., concentrations of PenStrep vs. antimycotic) for reproducibility.

  4. Clarify whether rings from the same donor were randomised into groups.

  5. Scale bars should be consistently reported in figure legends (H&E, Alcian Blue, TUNEL). Add representative negative controls in immunostaining images for clarity.

  6. A non-parametric Wilcoxon test is reported as “two-tailed t-test.” This is inaccurate. It should be corrected to “Wilcoxon matched-pairs signed rank test.”

  7. The limitations section should be expanded to include donor-to-donor variability, absence of dynamic mechanical cues (airflow, stretch), and absence of immune or vascular compartments.

Author Response

Comment 1. The authors should more explicitly distinguish their model from recent large airway explant cultures (Lee-Ferris et al., 2025) that also demonstrated viability up to 14 days. Current framing risks overstating novelty.

Answer 1. Thank you for the comment. We have added additional text in discussion to compare our work with Lee-Ferris et al explaining both the advantages and limitations of our work.

Changes made: We have added the following to the end of introduction [Line 98-110].

Our proposed model is designed specifically to capture the immediate cellular and molecular events relevant to early airway complications. This focus is highly justified as early epithelial death at the airway anastomosis site and the initial inflammatory cascade leading to stenosis are critical events occurring within the first post-injury or post-lung transplant. In contrast, recent protocols such as Lee-Ferris et al model are engineered for long-term viability and require advanced components like Gelfoam scaffolds and Air-Liquid Interface (ALI) maintenance. Our culture maintains the tissue in a state closest to its initial condition, which is crucial for accurately measuring mucus formation, basal metabolism, or inflammatory signaling that might be compromised or altered after two weeks of maintenance. By utilizing a simple, static culture medium and a shorter incubation period, our model is significantly more cost-effective and resource-efficient than the long-term Gelfoam/ALI-based protocols.

Comment 2. Suggest adding a comparative table summarizing existing models (ALI, PCLS, explants, EVLP) versus the proposed protocol.

Answer 2. Thank you for bringing this to our notice. We have added a table comparing these models with our proposed model.

Changes made: We have added this table to the introduction section. [Line 111]

Comment 3. The study only reports up to 3 days of culture, while earlier models showed extended viability. This limitation must be clearly acknowledged in the abstract and discussion, with justification for the chosen endpoint.

Answer 3. Thank you for the suggestion. The aim of our study was to understand the acute response in the donor lung tissue which usually occurs in the proximal bronchus. This response is triggered early and is into effect even before 24 hrs. In reality, the bronchial anastomosis site does not have any perfusion but only exposed to air/O2 by breathing. In the controls of this experiment, we used a little more support by giving media as a cell culture level, but it does not support the airway epithelial viability, suggesting the clinical airway anastomosis site has a more risk to disturb epithelial viability. We intend to develop a strategy to prevent severe airway epithelial necrosis leading to dehiscence, and this study tells us the scavenging of metabolic toxins may be a key. Thus, we focused on investigating the proximal tracheal cultures in this early duration while trying to keep the tissue as close to its original state.

Changes made: We have explained this rationale in abstract and discussion. [Line 316-320].

Our study was limited to only 72 hours compared to other specialized protocols that are carried out for up to 14-50 days. In contrast to these studies our aim was to study acute responses in the donor tissues as the EVLP for lung transplants are usually only required for this duration. Further, it is well known that the airway complications are triggered very early at the proximal bronchus junctions and take effect within first 72 hours.

Comment 4. Only three donor samples were used. This small number raises concerns about reproducibility and donor variability. The limitation is mentioned but needs stronger emphasis, ideally with data on inter-donor variability.

Answer 4. Thank you for pointing this out. We believe 3 patient data is sufficient to establish the reproducibility of the protocol in terms of its effectiveness compared to the control cultures. We have added supplementary figure 1 to show the data for individual patients which shows very low variability between different samples. The statistical tests also show significant difference in the viability of the optimized cultures.

Changes made: We have added additional supplementary figure 1 to show the low variability of different donors.

Comment 5. Beyond glycoprotein synthesis, functional assays (e.g., barrier integrity, cytokine release, ciliary motion) would strengthen the model’s translational value. The absence of such data should be discussed.

Answer 5. We have indeed provided discussion on these limitations in our previous version but have further expanded this to be more explicit.

Changes made: Discussion has been modified to expand on this limitation: [Please see Line 321-328].

Furthermore, our model also lacks analysis of functional properties of the cultures such as airflow, ciliary motion, and shear stress which reduces its physiological relevance, particularly in studies involving mucociliary clearance or epithelial barrier integrity. Future work could incorporate assays such as transepithelial electrical resistance (TEER) or paracellular flux measurements to gauge barrier integrity, ELISA or other methods to quantify the release of relevant cytokines and inflammatory mediators, and high-speed video microscopy to evaluate ciliary beat frequency (CBF) and overall ciliary motion.

Comment 6. The authors suggest use in drug testing and translational research, but with only short-term viability shown, this claim seems overstated. Recommend tempering claims or providing preliminary pilot data (e.g., exposure to therapeutic agents).

Answer 6. We agree that our current experiments do not provide any evidence to support these claims. We have removed the line stating these claims.

Changes made: We have removed these lines from the text.  

Minor Revisions

Comment 1. Revise phrasing in the abstract: “Control tissue exhibited significant acidosis…” should clarify whether this is compared to pre-culture baseline.

Answer 1. We have updated the line to mention that acidosis was observed in control culture after 3 days. [Line 21].

Changes made: Updated the line to mention that acidosis was observed in control culture after 3 days [Line 21].

Control tissue exhibited significant acidosis after 3 days.

Comment 2. Include explicit mention of study limitations (short viability, small sample size) in the abstract and introduction.

Answer 2. We have discussed the limitations of short duration of culture in the discussion. In terms of sample size, 3 patient data in addition to statistically significant results show that the sample size should be sufficient.

Changes made: We have discussed the limitations in the discussion section. (P7L309-321)

Comment 3. Provide more details on medium composition (e.g., concentrations of PenStrep vs. antimycotic) for reproducibility.

Answer 3. Thank you so much for mentioning this point. We have updated the method section to provide more details on the reagents used and composition [Lines 121-130].

Changes made: We have mentioned this in line 121-130.

The tissues were divided into two groups: 1) cultured in a 6 cm2 dish with 8 ml of medium (DMEM (with Bicarbonate) [Corning, cat# 10-013-CV] /5%FBS [Gemini Bio, cat# 100-106] /10 mM HEPES [Gibco cat# 15630080]/80 µL, 1X PenStrep [Gibco, cat# 15-140-122]) with daily media change (Control) and 2) cultured in a 25 cm2 flask with 20 ml of medium (DMEM (with Bicarbonate)/5%FBS/20 mM HEPES/250 µL, 1X Antibiotic-antimycotic [ThermoFischer Scientific, cat# 15240062]) with twice-daily media change (Optimized).

Comment 4. Clarify whether rings from the same donor were randomised into groups.

Answer 4. Yes, the rings used for control and optimized groups were selected randomly for each donor sample.

The rings were selected randomly for culture in a control or optimized model.

Changes made: We have added this to the Methods section [Line 120]

Comment 5. Scale bars should be consistently reported in figure legends (H&E, Alcian Blue, TUNEL).

Add representative negative controls in immunostaining images for clarity.

Answer 5. We are not sure what the reviewer means by this comment. We have added scale bars in each of the panel A, B and C and have mentioned the scale bar size in the legend for each section. We did not perform any immunostaining in this study.

Comment 6. A non-parametric Wilcoxon test is reported as “two-tailed t-test.” This is inaccurate. It should be corrected to “Wilcoxon matched-pairs signed rank test.”

Answer 6. Thank you for pointing this out and we apologize for this error. We have corrected the text.

Changes made: We have corrected this in [Line 178].

The results were expressed as mean ± standard deviation. Significance between means was determined using a non-parametric Wilcoxon matched-pairs signed rank test, with a p < 0.05 considered statistically significant in all experiments.

Comment 7. The limitations section should be expanded to include donor-to-donor variability, absence of dynamic mechanical cues (airflow, stretch), and absence of immune or vascular compartments.

Answer 7. We have already discussed limitations of donor-to-donor variability and its effects on scalability of the model in the initial version of the manuscript. We have further expanded the section to discuss the absence of mechanical cues and vascular compartments.

Changes made: The following text has been added to the discussion section: [Line 333-343]

The current model lacks dynamic mechanical cues such as the cyclic stretch and compression that occur with normal breathing, as well as the airflow and associated shear stress across the luminal surface. These mechanical forces are known to be critical regulators of epithelial cell function, including differentiation, barrier integrity, and ciliary beat frequency, and their absence may lead to non-physiological cellular responses or maintenance of phenotype. Furthermore, the model inherently omits the complexities of the systemic immune and vascular compartments. Crucially, the absence of circulating immune cells (e.g., T cells, B cells) and the local immune microenvironment means that our study cannot evaluate host-graft immunological interactions, the potential for allograft rejection or chimerism, and the full spectrum of inflammatory responses that would occur in vivo.

Round 2

Reviewer 2 Report

Comments and Suggestions for Authors

The authors have addressed most of the points carefully. I appreciate the authors for their clarification of experimental designing and presentation. By directing the application of proposed technique being more focused on specific situation, I believe that the significance of this work has become clearer. Following are further suggestions in the revisions that the authors made.

Major comments.

  1. Lines 97 – 100: this is overstating. The authors have not compared the proposed model to for example immediately isolated trachea or gel form-ALI model. They also do not have sufficient analysis confirming this statement (regarding to mucus production, metabolism, inflammation states). They also do not include any data from immunofluorescent microscopy analysis, transcriptome, etc, that are commonly used in the field to confirm how close the epithelia are to initial state and its integrity. I suggest either completely remove this statement or discuss as future direction to assess how well this model preserves initial state of proximal airway and list specific assessments that should be performed.
  2. Table 1, proposed model, cellular architecture: the same reason as above, mentioning “integrity” here is overstating.
  3. Table 1, ALI, cellular architecture: this model’s cellular architecture and cell composition are culture media dependent.

Author Response

C1. Lines 97 – 100: this is overstating. The authors have not compared the proposed model to for example immediately isolated trachea or gel form-ALI model. They also do not have sufficient analysis confirming this statement (regarding to mucus production, metabolism, inflammation states). They also do not include any data from immunofluorescent microscopy analysis, transcriptome, etc, that are commonly used in the field to confirm how close the epithelia are to initial state and its integrity. I suggest either completely remove this statement or discuss as future direction to assess how well this model preserves initial state of proximal airway and list specific assessments that should be performed.

Answer: We removed sentences L97-100 in the revised manuscript.

C2. Table 1, proposed model, cellular architecture: the same reason as above, mentioning “integrity” here is overstating.

C3.Table 1, ALI, cellular architecture: this model’s cellular architecture and cell composition are culture media dependent.

Answer for C2&3. We removed the row, Cellular architecture, from Table 1.

Reviewer 3 Report

Comments and Suggestions for Authors

The manuscript seems ready for publication at the moment. All points have been clarified. 

Author Response

We appreciate your endorsement for the publication of our manuscript.